# High-Density Lipoprotein Alterations in Type 2 Diabetes and Obesity

**DOI:** 10.3390/metabo13020253

**Published:** 2023-02-09

**Authors:** Damien Denimal, Serge Monier, Benjamin Bouillet, Bruno Vergès, Laurence Duvillard

**Affiliations:** 1INSERM, UMR1231, University of Burgundy, 21000 Dijon, France; 2Department of Biochemistry, CHU Dijon Bourgogne, 21000 Dijon, France; 3Department of Endocrinology and Diabetology, CHU Dijon Bourgogne, 21000 Dijon, France

**Keywords:** high-density lipoprotein, type 2 diabetes, obesity, metabolic syndrome, cholesterol efflux, glycoxidation, endothelium

## Abstract

Alterations affecting high-density lipoproteins (HDLs) are one of the various abnormalities observed in dyslipidemia in type 2 diabetes mellitus (T2DM) and obesity. Kinetic studies have demonstrated that the catabolism of HDL particles is accelerated. Both the size and the lipidome and proteome of HDL particles are significantly modified, which likely contributes to some of the functional defects of HDLs. Studies on cholesterol efflux capacity have yielded heterogeneous results, ranging from a defect to an improvement. Several studies indicate that HDLs are less able to inhibit the nuclear factor kappa-B (NF-κB) proinflammatory pathway, and subsequently, the adhesion of monocytes on endothelium and their recruitment into the subendothelial space. In addition, the antioxidative function of HDL particles is diminished, thus facilitating the deleterious effects of oxidized low-density lipoproteins on vasculature. Lastly, the HDL-induced activation of endothelial nitric oxide synthase is less effective in T2DM and metabolic syndrome, contributing to several HDL functional defects, such as an impaired capacity to promote vasodilatation and endothelium repair, and difficulty counteracting the production of reactive oxygen species and inflammation.

## 1. Introduction

Type 2 diabetes mellitus (T2DM) has reached epidemic proportions. More than 500 million adults worldwide live with diabetes, and patients with T2DM represent more than 90% of them [1]. The risk of developing T2DM depends on a combination of genetic (polygenic) and environmental factors. T2DM has been more frequently associated with age, obesity, family history of T2DM, history of gestational diabetes, physical inactivity, and certain ethnicities. Some biomarkers can predict the development of T2DM. Thus, 403 distinct association signals with T2DM were identified in a large meta-analysis of genome-wide association studies [2]. Several non-genetic biomarkers, such as metabolites or lipids for instance, are also able to predict the development of T2DM (reviewed in [3]).

The risk of fatal and non-fatal cardiovascular disease (CVD) is significantly increased in patients with T2DM [4], obesity [5], or metabolic syndrome (MetS) [6]. The progression to cardiovascular (CV) complications involves many complex and often overlapping mechanisms, such as insulin resistance, oxidative stress, low-grade inflammation, endothelial dysfunction, nephropathy, and dyslipidemia. Typically, the dyslipidemia observed in insulin-resistant states is characterized by an elevated triglyceride (TG) level, the predominance of small, dense, low-density lipoproteins (LDLs), and a low high-density lipoprotein cholesterol (HDL-C) level [7]. Hypertriglyceridemia is due to increased production and decreased catabolism of very low-density lipoproteins (VLDLs). An excellent review in the first volume of this special issue of *Metabolites* focused on structural and functional changes in LDL particles in T2DM [8]. Besides the low plasma HDL-C level, HDLs are also qualitatively, kinetically, and functionally altered in T2DM and obesity. The aim of the present review is to discuss the main findings regarding HDL alterations in T2DM and obesity in the context of CVD.

Regarding HDLs, it is well established in large epidemiological studies that low HDL-C level is associated with an increased risk of CVD [9,10]. However, the relationship between HDL-C and CVD is more complex [11], and HDL-C is not a reliable marker of CV risk in specific populations. In fact, extensive evidence suggests a U-shaped relationship between HDL-C concentration and CV events or mortality [12,13,14]. In addition, studies on genetic polymorphisms associated with low or high HDL-C levels have often failed to demonstrate a causal relationship between HDL-C level and CV risk [15,16,17]. This highlights the limitations of HDL-C concentration as a universal biomarker of CV risk. Research subsequently turned to the study of the antiatherogenic functions of HDLs as a better biomarker of CV risk than HDL-C alone.

## 2. HDL Size and Composition

HDLs are composed of apolipoproteins (mainly apoA-I), non-structural proteins, and lipids. They are a heterogeneous group of particles that vary in size as well as protein and lipid composition. Basically, HDLs are composed of a surface monolayer, made up of amphipathic phospholipids (PLs) and sphingolipids (SPLs) and unesterified cholesterol (UC) molecules, which surround the particle core, which consists of a mixture of hydrophobic TG and cholesteryl ester (CE) molecules. The alterations in HDL size and composition in contexts of insulin resistance are important because there is a close relationship between HDL size and composition and their functions. An overview of the changes in HDL size and composition is presented in Figure 1.

### 2.1. Particle Size

Size-based HDL nomenclature can be confusing because it depends on the technology used to separate HDL subclasses. In any case, HDLs are classified as very small, small, medium, large and very large particles. Classically, those particles are called HDL2a and HDL2b (large), HDL3a, HDL3b, and HDL3c (medium or small) and unlipidated apoA-I or pre-β (discoidal).

Overall, the size of HDLs is decreased in T2DM [18,19]. More precisely, nuclear magnetic resonance spectroscopy studies revealed that small HDL particles are more numerous in T2DM, contrary to large HDL particles [20,21,22,23]. It is also the case in insulin-resistant individuals without diabetes [23,24,25], and the intensity of changes actually increases with the degree of insulin resistance [23]. Consistent with these observations, HDL2-C has been reported to be inversely associated with insulin resistance and T2DM in a large cohort of community residents, whereas HDL3-C had the opposite association [26]. The shift in HDL population towards smaller HDL particles is most likely linked to a modified metabolism. More precisely, the higher CE/TG exchange between TG-rich lipoproteins and HDLs in T2DM leads to the depletion in CEs of HDL particles, accompanied by an enrichment in TGs [27,28,29,30]. This higher CE/TG exchange is due to the upregulation of CETP activity [19,27,31]. The impairment of the inhibitory function of apoC1 on CETP activity likely contributes to the increase in CETP activity [31,32]. It has been suggested that such a remodeling of the core of HDL particles renders them more prone to endothelial and hepatic lipases, two enzymes hydrolyzing TGs and PLs, that likely leads to smaller HDL particles. In fact, endothelial lipase level is increased in T2DM [33,34] and increases with body mass index (BMI) [35].

Changes in pre-β HDL level in diabetes and obesity are more controversial. Some studies found it to be decreased in T2DM [27,36], correlating with the severity of nephropathy [37]. Others observed no changes [28], or even found elevated concentrations [38] of pre-β HDL levels in T2DM and MetS.

### 2.2. Lipidome

As mentioned above, HDL particles are enriched in TGs in patients with T2DM [18,19,39,40,41], obesity [42], insulin resistance [40,43] or MetS [44]. The replacement of CE by TG molecules in HDLs affects the conformation of apoA-I [45], which could modulate binding to receptors and the functions of HDLs.

PLs and SPLs represent more than one third of the total mass of HDLs. They play a major role in HDL functions, either by binding to a specific receptor, such as sphingosine-1-phosphate (S1P) receptors or by modulating the physicochemical properties of HDLs. Lipidomic studies have revealed several changes in the HDL phosphosphingolipidome, although the results are quite heterogeneous. The expression of results in relation to either total HDL mass, total HDL lipids, or apoA-I levels makes it challenging to compare between studies. The HDL content in total PLs has been found to be either decreased in T2DM when the results are expressed in relation to total lipids [40], or normal when the results are expressed in relation to total mass or total lipids [18,41].

Taken together, phosphatidylcholines (PCs, the main PL class) and sphingomyelins (SMs) represent more than 80% of PLs/SPLs in HDLs. When the results are expressed in relation to apoA-I, the HDL content in PCs and SMs in T2DM has been found to be either normal or decreased [19,41]. When the results are expressed in relation to total HDL lipids, PCs and SMs have been found decreased in T2DM HDLs [40]. In addition, Cardner et al. reported a decrease in PC36:2 and 34:2, but the way in which results are expressed is unclear [22]. The SM/PC ratio, a marker of surface rigidity, has been found to be either unaffected or decreased [19,40].

HDL-lysoPCs were most often reported as increased in T2DM [19,40], prediabetes [40], and MetS [46], although Cardner et al. recently found a decrease in T2DM [22]. LysoPCs in HDLs mainly derived from the hydrolysis of PCs by lecithin:cholesterol acyltransferase (LCAT) and lipoprotein-associated phospholipase A2 (Lp-PLA2). Lp-PLA2 is more active in T2DM [47,48], and Lp-PLA2 is positively associated with obesity [48]. Data on LCAT are scare, showing an elevation in only two studies [49,50], making it difficult to understand the precise role of LCAT in the production of HDL-lysoPCs in diabetes and MetS.

HDLs are enriched in many phosphatidylethanolamine (PE) species in T2DM [22,41] and MetS [46]. Although the underlying mechanism is not yet understood, such a change could be relevant to note since an enrichment of reconstituted HDLs (rHDLs) with PE36:5 reportedly impairs their antiatherogenic properties [51]. Moreover, HDL2-PE36:5 is positively associated with atherosclerotic plaques [51].

In addition, HDL-plasmalogens are decreased in T2DM [19,22,41] and in non-diabetic MetS individuals [46]. Acting as scavengers to remove oxygen radicals in HDLs, plasmalogens may be overconsumed due to increased oxidation of HDLs in diabetes. Such a decrease in HDL-plasmalogens is worth noticing because of the role of these lipids in HDL-mediated antioxidant and antiapoptotic properties. Interestingly, it has been observed that reduced HDL-plasmalogen levels is associated with coronary artery disease (CAD) [52].

Last, but not least, the circulating concentration of S1P, three-quarters of which are carried by HDLs, was often found to be decreased in T2DM [53,54,55]. In addition, the plasma levels of apoM, the carrier protein of S1P in HDLs and almost entirely found in HDLs in the blood, are decreased in T2DM individuals [55,56,57,58]. However, the S1P content in HDL particles is most often normal in T2DM [41,59,60]. Surprisingly, the S1P content in HDLs is decreased in obese patients without diabetes [61,62]. Such a decrease is of particular interest due to the important role of this lipid for HDL functions; HDL-S1P level is associated with CV outcomes [63]. The reason for the difference between diabetic and non-diabetic obese patients is not well understood, very similar to the link between diabetes/obesity and S1P metabolism. However, it has been shown that insulin-resistant obese mice have reduced apoM gene expression and HDL-S1P levels [62]. Moreover, hepatic apoM mRNA levels decrease in hyperglycemic conditions in rats and in hepatocytes [64]. Although it has been shown that in vitro glycation of HDLs impairs the binding of S1P to HDL particles and that this accelerates S1P catabolism [65], this is inconsistent with the normal S1P level observed in T2DM HDLs [41,59,60]. Interestingly, it has been recently demonstrated that the hepatic transcription factors forkhead box O (FoxO), which are critical mediators of the hepatic insulin signaling pathway, promote apoM expression, and drive S1P content in HDLs [62]. A specific decrease in lipoprotein-related FoxO targets is observed in obese mice, although the molecular basis remains unknown to date [62].

### 2.3. Proteome

Proteins form a major structural and functional component of HDL particles. The protein cargo of HDLs is comprised of over 100 proteins [66]. It should be noted that the HDL isolation method has a significant impact on HDL protein composition [67], which could contribute to some heterogeneity among studies.

The HDL proteome has been extensively studied in diabetes, especially using mass spectrometry. Besides apoA-I, the kinetics of several HDL-proteins is perturbed in T2DM, as described hereafter. Reduced half-lives are observed for apoA-II, apoJ, apoA-IV, transthyretin, vitamin-D binding protein, and complement C3 [66]. A recent large proteomic study evaluated 182 proteins in isolated HDL fraction and found that HDLs from T2DM patients are enriched in 17 proteins and are deprived of 44 proteins compared to healthy individuals [22]. More precisely, HDLs were particularly enriched in pulmonary surfactant protein B and in serum amyloid A proteins (SAA1 and SAA2). In contrast, T2DM HDLs were deprived in apoA-IV, clusterin, paraoxonases (PON1 and PON3), apoD, apoE, apoF, apoM, apoC-II, and apoC-III [22]. The enrichment of HDLs in SAA in diabetes [22,60,68] is of particular interest due to its deleterious role in reverse cholesterol transport (RCT) [68,69]. The lower content of HDLs in paraoxonases likely contributes to the impaired antioxidant function, which we will discuss below. Moreover, the decrease in apoA-IV content in HDLs may be also of note. Indeed, a low plasma level of apoA-IV is associated with CAD [70], and apoA-IV is known to participate in a broad spectrum of biological processes, including RCT and more generally protection against atherosclerosis (reviewed in [71]). Lastly, as mentioned above, apoM levels are decreased in the plasma of T2DM patients [55,56,57,58].

### 2.4. Glycation and Oxidation

Chronic hyperglycemia in diabetes facilitates accelerated glycation, which is usually divided into early and late glycation. Thus, reducing sugar reacts with amino groups of proteins to form Schiff bases and then Amadori compounds (early glycation products). It may be followed by irreversible dehydration, condensation, and cross-linking reactions, resulting in a heterogeneous family of derivatives called advanced glycation end-products (AGEs), also known as late glycation products or glycoxidation products. Besides glucose itself, chronic hyperglycemia and oxidative stress in diabetes induces the production of dicarbonyls (glyoxal, methylglyoxal, 3-deoxyglucosone), which also react with proteins to yield AGEs [72]. Carboxymethyl-lysine (CML) is thought to be the most abundant AGE in vivo. Among the amino acids with nucleophilic residues prone to glycation, lysine residues are particularly abundant in apolipoproteins, and they are the preferred site of glycation.

The level of HDL glycation is significantly increased in T2DM, and seven glycation sites on lysine residues of apoA-I have been identified in T2DM subjects [73]. T2DM HDLs are enriched in methylglyoxal [74]. However, Low et al. observed a similar CML level in HDLs from patients with T2DM and non-diabetic individuals [27].

Several in vitro studies suggest that the glycation or glycoxidation of HDLs could modulate some properties of these particles. For instance, the glycation of HDLs by methylglyoxal induces structural alterations of HDL particles, decreasing their stability and plasma half-life [74]. This also decreases PON1 activity [75]. In addition, advanced glycation of apoA-I or HDLs severely impaired multiple functions, including cholesterol efflux [76], stabilization of ATP-binding cassette (ABC)A1 [76], and expression of adhesion molecules [76]. The glycation of HDLs also resulted in an impaired binding of S1P to HDLs, and S1P on glycated HDLs degrades faster [65].

Beyond its role in the glycoxidation of HDL proteins, oxidative stress also leads to the peroxidation of lipids in HDLs. The proinflammatory enzyme myeloperoxidase (MPO) represents a major source of reactive-oxygen species (ROS) in atherosclerotic lesions. MPO-derived oxidants modify lipids and lead to the formation of dicarbonyls, such as malondialdehyde (MDA). The increased oxidative stress in subjects with T2DM and/or obesity is a consequence of several abnormalities, including hyperglycemia, insulin resistance, inflammation, and dyslipidemia [77]. Thus, not surprisingly, T2DM patients have increased concentrations of oxidized HDLs [78] and apoA-I [79], and their HDLs are enriched in MDA [80] and oxidized fatty acids derived from arachidonic and linoleic acids [81]. A positive feedback loop is set up, since, by binding to the lectin-like oxidized LDL (LOX-1) receptor at the plasma membrane of endothelial cells, oxidized HDLs trigger an increased expression of NADPH oxidase-2 (another major source of ROS in atherosclerotic lesions), tumor necrosis factor (TNF)-α, and LOX-1 receptor [82]. Lastly, the oxidation of HDLs with hypochlorite, which is produced by MPO, leads to the binding of HDLs to CD36 on platelets, thus triggering platelet aggregation [83].

However, the conclusions of such in vitro experiments should be taken with caution considering that in vitro modified HDLs may be differentially or more extensively altered than those occurring in vivo. Glucose or reactive intermediate (e.g., methylglyoxal or hypochlorite) concentrations and incubation times used in vitro are sometimes well beyond what occurs naturally in humans.

### 2.5. Carbamylation

Carbamylation (carbamoylation stricto sensu) is another post-translational modification affecting lipoproteins in diabetes. It is an irreversible non-enzymatic process mediated by isocyanate, and it is characterized by the binding of a carbamoyl moiety (-CONH_2_) to lysine, resulting in carbamyl-lysine. It originates from MPO within atherosclerotic lesions or from urea or smoking. Plasma MPO levels are increased in T2DM [84,85], and patients with T2DM exhibit higher levels of carbamylated HDLs even without renal impairment [86,87]. Interestingly, plasma levels of carbamylated proteins have been shown to predict the risk of major adverse cardiac events [88]. However, above all, a recent prospective study has shown that carbamylated HDL serum levels in T2DM patients are independently associated with all-cause and cardiovascular-related mortality [89]. This relationship may be causative since some of the atheroprotective properties of HDLs are altered after carbamylation (Figure 2). Thus, carbamylated HDLs are less effective at removing cholesterol from macrophages [90,91], inhibiting monocyte adhesion and recruitment [87,92], and protecting LDLs from oxidation [93].

## 3. HDL/apoA-I Kinetics

Numerous in vivo kinetic studies demonstrate that low HDL-C levels in T2DM and obesity are due to an accelerated catabolism of HDL particles [43,44,94,95,96]. The fractional catabolic rate (FCR) of HDL-apoA-I is increased, resulting in shorter plasma residence times for HDL particles [39,94,97]. Obese patients at an early stage of insulin resistance (i.e., without impaired glucose tolerance) already have an accelerated HDL-apoA-I catabolism [98].

Studies have found the production rate (PR) of HDL-apoA-I to be either elevated [39,43,95,97] or normal [44,98] in insulin-resistant individuals. The PR of HDL-cholesterol has recently been reported to be higher in patients with T2DM. In any case, the increase in HDL-apoA-I PR is usually smaller than the increase in the FCR, which explains the decreased concentrations of HDL-apoA-I. Interestingly, chronic endogenous hyperinsulinemia without insulin resistance (patients with insulinoma) does not induce an increase in the HDL-apoA-I PR [43]. This suggests that hyperinsulinemia per se is not responsible for the increased PR of HDL-apoA-I that is sometimes reported in patients with insulin resistance.

Glycation and glycoxidation may play a role in the accelerated catabolism of HDLs, although there is reportedly no correlation between HDL-apoA-I FCR and HbA_1c_ [96]. Thus, the turnover of glycated apoA-I is almost three times faster than its non-glycated counterpart [97]. In addition, the methylglyoxal modification of HDLs reduces plasma half-life in vivo [74]. Yet considerable evidence suggests that the kinetic properties of HDLs are linked to TG metabolism. For instance, the HDL-apoA-I FCR in obese individuals is associated with triglyceridemia [99,100], VLDL-apoB PR [99], VLDL_1_-TG FCR and PR [100], and also with the HDL-TG/CE ratio [98]. In T2DM, the HDL-apoA-I FCR is also positively associated with plasma TG [96] and with the TG content in HDLs [44,96]. HDLs enriched in TGs are hydrolyzed by hepatic lipase, leading to lipid-poor HDLs. Such HDLs are thermodynamically unstable and exhibit structural modifications in apoA-I, which facilitates both its dissociation from HDL particles [101] and its renal glomerular filtration [102]. Moreover, in vivo TG enrichment of HDLs using a TG emulsion resulted in a 26% increase in the FCR of HDL-apoA-I [103]. Lastly, the FCR of HDL-apoA-II, the second major apolipoprotein in HDLs, is independently associated with VLDL_1_-TG FCR in obese patients [104].

In addition to hepatic lipase, endothelial lipase is major determinant of HDL-C levels. High endothelial lipase serum concentrations have been observed in T2DM patients [33,34], and angiopoietin-like protein 3 (ANGPTL3), which is an inhibitor of endothelial lipase, is decreased in T2DM patients [60]. Therefore, the increase in endothelial lipase activity likely contributes to the increased HDL catabolism in these patients.

## 4. HDL Functions

### 4.1. Reverse Cholesterol Transport

The best-known function of HDLs is their major role in RCT, which enables the removal of cholesterol from lipid-laden macrophages and artery walls. To sum up, HDLs promote cholesterol efflux from macrophage foam cells in atherosclerotic plaques either specifically by interacting with the transporters ABCA1 and ABCG1, or by aqueous diffusion, through a process facilitated by the scavenger receptor B1 (SR-B1). ABCA1 mediates cholesterol efflux preferentially to lipid-poor apoA-I and small dense HDLs, whereas ABCG1 transports cholesterol to the more mature lipidated HDL particles. Free cholesterol is then esterified by LCAT, and this esterification is important for maintaining the dynamics of cholesterol efflux. Cholesterol is ultimately cleared by the liver, either directly by selective uptake through SR-B1 or by a more recently discovered pathway mediated by F1-ATPase and P2Y13 receptor, or indirectly after CETP-mediated transfer to apoB-containing lipoproteins, which are then internalized by the LDL receptor. In addition to this classical hepatobiliary pathway, cholesterol can be also eliminated by a transintestinal pathway called transintestinal cholesterol efflux (TICE) [105].

Cholesterol efflux is thus the first step in the atheroprotective RCT pathway, and the cholesterol efflux capacity (CEC) of HDL particles is a crucial determinant of cholesterol clearance from lipid-laden macrophages. Over the past few years, studies have demonstrated that the CEC of HDLs is more strongly and inversely associated with incident cardiovascular events than the circulating HDL-C level itself [106,107]. For instance, it has been reported that HDL CEC is inversely associated with incident coronary heart disease events independently of traditional cardiovascular risk factors, HDL-C, and apoA-I levels [108]. Interestingly, HDL CEC was inversely correlated with T2DM and measures of obesity in this large study [108].

The studies that measured CEC in T2DM or obese patients yielded contrasted results. The causes of this heterogeneity are not clearly understood, but it should be noted that there is currently no gold standard for assessing CEC. It is therefore evaluated using different protocols in terms of cell type (macrophages, fibroblasts, hepatocytes, transfected cells), the induction or not of cholesterol transporters, the tracer (radioactive or fluorescent), the acceptor (HDLs, whole serum, apoB-depleted serum), its concentration, and incubation time. These differences often make the comparison between studies difficult and likely explain the divergent results. As an illustration, whole serum CEC using human THP-1 macrophages was a strong predictor of survival in a large cohort of patients with acute myocardial infarction, but this association was lost in the same cohort using the combination of apoB-depleted serum with THP-1 cells or the combination of whole serum with ABCA1-overexpressing CHO cells or Fu5AH cells [109]. In addition, a recent paper found that the HDL isolation procedure clearly modulates CEC results [67].

Table 1 summarizes the most relevant CEC studies in T2DM and/or MetS. Some authors reported an increased CEC in patients with T2DM [27] and in diabetic patients with hypertriglyceridemia [110]. ABCA1-dependent efflux was also increased using apoB-depleted serum from T2DM patients with hypertriglyceridemia compared to T2DM patients without hypertriglyceridemia [111]. On the other hand, CEC in T2DM patients was unmodified using fibroblasts and whole plasma [30,55,110], human THP-1 macrophages, and apoB-depleted serum [41], or when using THP-1 cells and HDLs isolated by dextran sulfate precipitation [81]. Lastly, CEC was decreased in T2DM patients using adipocytes and LpA-I (i.e., HDL particles containing apoA-I but not apoA-II) [112], Fu5AH hepatoma cells and whole plasma/serum [34,36,37,68,113,114], mouse peritoneal macrophages and isolated HDL3 [115], murine RAW264.7 macrophages and apoB-depleted serum [97], and also THP-1 macrophages and isolated HDLs [18,116]. Similarly, it was recently reported that small HDL particles and apoB-depleted serum from patients with T2DM both have impaired ABCA1-dependent CEC using baby hamster kidney (BHK) cells [97,117]. However, medium and large HDL particles had a similar capacity to promote ABCA1-specific CEC in T2DM patients compared to control individuals in this study [117]. Otherwise, all three sizes of HDL particles from T2DM subjects had similar ABCG1-dependent CEC compared to controls [117].

Several studies have tried to determine the potential variables of CEC in T2DM and obesity. Regarding HDL particle size, ABCA1-dependent efflux typically correlates with lipid-poor pre-β HDLs, but not with large HDL particles, and it has been shown that CEC correlates with pre-β HDL levels in T2DM [36,37]. Moreover, a positive association between HDL particle size and CEC was observed in obese adolescents [118] and in a general adult population [119].

The impact of just glycation of HDLs on CEC is a matter of debate. While the glycation of HDLs did not alter CEC in murine macrophages in several studies [115,123,124], it has also been found that glycated HDL3s promote cholesterol efflux less efficiently [125], and the decreased CEC in T2DM patients is associated with the glycation level of lysine residues in HDLs [97]. Nevertheless, glycoxidative modifications may play a greater role, although CML levels do not appear to be associated with CEC in diabetic patients [27]. For instance, serum AGEs is inversely correlated with CEC in T2DM patients with nephropathy [37]. However, above all, CEC diminishes after glycoxidation of HDLs by 3-deoxyglucosone [126], glycolaldehyde or MDA [123], or glucose and copper [124]. Similarly, the modification of rHDLs by methylglyoxal or glycolaldehyde alter CEC [127], and AGE-modified apoA-I is less effective at promoting cholesterol efflux than unmodified apoA-I [76,128,129]. Treatment of mice with a scavenger of dicarbonyls, such as MDA, was found to ameliorate HDL CEC [129]. The deleterious effect of glycoxidation on HDL CEC could be mediated by the inactivation of LCAT. Indeed, HDL3 oxidized by MPO [130] and rHDLs modified by methylglyoxal [131] lose their LCAT activity, which is important for maintaining the dynamics of cholesterol efflux. In addition, CEC positively correlates with LCAT activity in T2DM [30].

Annema et al. reported that CEC is negatively associated with a low-grade inflammation score in MetS individuals [121]. Inflammation can alter the HDL composition in proteins and lipids and can also change levels of CETP or phospholipase A2 [132]. It is well established that inflammatory remodeling of HDLs impairs CEC [133]. For instance, the level of SAA, an acute-phase response protein of inflammation, is increased in T2DM HDLs, and CEC negatively correlates with SAA levels in serum [68] and in HDLs [133]. Interestingly, HDLs in SAA-deficient mice with inflammation are protected from loss of CEC [133]. Nevertheless, it is not well established whether the modestly elevated level of SAA found in T2DM HDLs in a context of low-grade inflammation is sufficient to significantly affect CEC.

He et al. recently identified serpin family A member 1 (SERPINA1), a protease inhibitor also known as alpha-1-antitrypsin, as a potential contributor to the altered CEC in diabetic patients [117]. Small HDLs from patients with T2DM are deprived in SERPINA1, and reconstituting small HDLs with SERPINA1 improves CEC [117].

Changes in lipid composition may also affect HDL CEC. In particular, the replacement of CE by TG molecules in HDLs affects the conformation of apoA-I [45] and could therefore modulate binding to receptors. In addition, the literature suggests that the content of HDLs in total PLs [134,135] and in SMs [136] modulates CEC, but, as mentioned above, the changes in these parameters are very heterogeneous across studies. Cardner et al. recently reported that CEC of apoB-depleted serum is mainly driven by apoA-I level in diabetic individuals [22]. Nevertheless, our group recently reported that changes in HDLs observed in poorly-controlled patients with T2DM, such as an enrichment of 68% in TGs or of 41% in PEs, do not alter CEC, at least using a model with THP-1 cells and apoB-depleted plasma [41].

### 4.2. Anti-Inflammatory Properties

Both diabetes and obesity are associated with low-grade inflammation, which substantially contributes to endothelial dysfunction and atherosclerosis. As shown in Figure 3, HDL particles exert an anti-inflammatory function by downregulating the expression of molecules involved in the recruitment of immune cells into the subendothelial space. These molecules include chemokine CCL-2 [137], vascular cell adhesion molecule (VCAM)-1, intracellular adhesion molecule (ICAM)-1, and selectin-E [41]. In addition, HDLs inhibit the release of inflammatory cytokines, such as TNF-α and IL-1β. The HDL anti-inflammatory function seems of particular relevance for CV outcomes, since it predicts new cardiac events in patients with myocardial infarction, independently of HDL-C [138]. Moreover, an inverse association between the anti-inflammatory capacity of HDLs and incident CV events was recently observed in a study of individuals from the general population cohort, independently of both HDL-C and CEC [139].

Table 2 summarizes the studies on anti-inflammatory properties of HDLs in T2DM. HDLs from T2DM patients are less able to inhibit the migration of monocytes towards endothelial cells [87,140]. Interestingly, this loss of function correlates with plasma SAA [140] and carbamylated HDL levels [87]. The fact that in vitro carbamylation of HDLs reproduces the loss of capacity to inhibit the migration of monocytes reinforces the potential role of carbamylated HDLs [87]. Although HbA_1c_ or glucose levels do not correlate with this loss of HDL function [140], some evidence suggests that glycoxidative changes in HDLs may play a role. Thus, the ex vivo treatment of plasma with L-4F, an apoA-I mimetic peptide able to bind oxidized lipids with a higher affinity than apoA-I itself [141], restores the anti-inflammatory function of HDLs [140]. Moreover, AGE-modified apoA-I is less effective than apoA-I at inhibiting the expression of CD11b, an integrin located on monocytes that interacts with the adhesion molecule ICAM-1 [76].

As mentioned above, HDL particles are also able to downregulate the expression of adhesion molecules, such as ICAM-1 and VCAM-1, on endothelial cells [41], thus inhibiting the recruitment of immune cells into the subendothelial space. Dullaart’s group showed that HDLs from T2DM or MetS patients are less able to inhibit VCAM-1 gene expression than are those from healthy individuals [142,143]. This result correlated positively with plasma glucose and negatively with PON1 activity, suggesting a potential role of glycoxidative modifications. In vitro carbamylation of HDLs also leads to the same functional defect [87]. On the contrary, our group recently reported that the ability of HDLs to inhibit the expression of adhesion molecules is not impacted in patients with T2DM, despite classical alterations in HDL lipid composition [41]. HDL-associated apoM delivers S1P to S1P receptors (S1PR), and the HDL/apoM/S1P pathway is particularly effective at inhibiting the expression of adhesion molecules [144]. However, although apoM seems decreased in T2DM HDLs [22], several groups found a conserved level of S1P in T2DM HDLs [41,59,60], making the apoM/S1P axis unlikely to be responsible for this defect when it is observed. To conclude, it is worth noting that the infusion of rHDLs in T2DM patients significantly enhances the ex vivo ability of HDLs to inhibit the expression of adhesion molecules on endothelial cells and to decrease the expression of CD11b on leukocytes [145]. The extent of this benefit is relative to the increase in the plasma concentration of total HDLs [145], suggesting that the HDL particle number is an important determinant of this anti-inflammatory HDL function in T2DM.

Otherwise, apoA-I [73] or HDLs [146] isolated from T2DM patients lose their ability to inhibit the release of TNF-α and IL-1β by macrophages after lipopolysaccharide stimulation. Interestingly, this is mimicked by in vitro glycation of apoA-I [73] or HDLs [146]. The decrease in the SM content of HDLs, observed by some groups [19,22,40], but not by others [41], could also contribute to this defect in HDLs. Indeed, rHDLs containing SMs are more able than rHDLs containing PCs to inhibit the release of proinflammatory cytokines [147].

The activation of the classical IKK/IκB-α/NF-κB pathway plays a crucial role in the expression of adhesion molecules and in the release of proinflammatory cytokines into cardiovascular tissue. HDLs/apoA-I inhibit the NF-κB pathway via several overlapped mechanisms following interaction with SR-B1, S1PR, ABCA1, and ABCG1: cholesterol efflux, endothelial nitric oxide synthase (eNOS) activation [148] and subsequent S-nitrosylation [149], upregulation of 3β-hydroxysteroid-δ24 reductase [150,151] and heme oxygenase-1 [151], and inhibition of the toll-like receptors TLR4 [152] and TLR2 [153]. HDLs or apoA-I ultimately prevent NF-κB p65 subunit translocation to the nucleus and DNA binding [152], thus inhibiting the transcription of adhesion molecules, CCL-2, proinflammatory cytokines, and also NADPH-oxidase (NOX2) genes. It has been shown that HDLs isolated from T2DM patients are unable to suppress the activating phosphorylation of the NF-κB p65 subunit in endothelial cells [54]. Glycated apoA-I partially loses its ability to inhibit cytosolic IκB-α phosphorylation and NF-κB p65 subunit translocation to the nucleus [73]. Since glycated apoA-I has a lower affinity to macrophages than native apoA-I [73], the lack of binding to receptors of glycated HDLs could be an explanation. In addition, carbamylated HDLs also increase the phosphorylation of p65 in endothelial cells, thus facilitating NF-κB p65 subunit translocation to nuclei [87].

### 4.3. Antioxidative Properties

T2DM and obesity are known to be associated with increased oxidative stress, which is closely linked to low-grade inflammation. In particular, oxidative stress in artery walls is responsible for the formation of oxidized LDLs (oxLDLs), rendering them more atherogenic. There are several mechanisms through which HDLs present in the intima can protect LDLs against oxidation [154]. HDLs directly protect LDLs from oxidation induced by one-electron oxidants (free radicals), and they are also able to remove oxidized lipids from LDLs. These activities can decrease local concentrations of oxLDLs. The antioxidative potential of HDL particles originates both from the activities of their proteins and from lipid components. Different HDL-associated apolipoproteins, lipid transfer proteins, and enzymes have been shown to contribute to the antioxidative capacity of HDLs. For instance, methionine residues located at positions 112 and 148 of apoA-I can reduce lipid hydroperoxides. ApoM binds oxidized phospholipids and enhances antioxidative activity of HDL [155]. In addition, PON1, which is exclusively associated with HDLs in the circulation, hydrolyzes PC-based oxidized PLs.

In vitro studies had demonstrated that HDLs from patients with T2DM have a decreased ability to metabolize lipid hydroperoxides [156]. The inhibitory effects of HDL3 on LDL oxidation is lower in patients with T2DM [157] or MetS [158]. However, this alteration is not observed when the antioxidative activity of total HDLs is analyzed [159]. In addition, HDLs from diabetic patients lose their inhibitory effect on endothelial superoxide production or NADPH oxidase activity [80]. A recent study showed that HDLs induce a higher level of ROS in aortic endothelial cells in T2DM patients compared to controls [81]. Moreover, it is well documented that PON1 activity is diminished in T2DM [49,97,116,142,156].

Glycoxidative modifications in HDLs again appear to play a significant role, since glycation by methylglyoxal decreases PON1 activity [75]. In addition, glycated HDLs are unable to counteract the inhibitory effect of oxidized LDLs on endothelium-dependent vasorelaxation [160], which could be due, at least in part, to a reduction in the antioxidant capacity of glycated HDLs.

Changes in antioxidative HDL functions could be also due to altered lipid composition. Several studies reported that TG enrichment and CE depletion in HDL3 subfractions are associated with diminished antioxidative activity [157,158,161]. The replacement of CEs by TGs in the HDL lipid core affects the conformation of apoA-I [45,101], thereby potentially modifying the accessibility of methionine residues to lipid hydroperoxides. Otherwise, the depletion of plasmalogens in HDLs likely contributes to reduced antioxidant capacity in T2DM and MetS due to their ability to remove oxygen radicals. Lastly, it was recently shown that high levels of oxidized fatty acids in T2DM HDLs impair their ability to inhibit ROS production [81]. Interestingly, the ex vivo treatment of T2DM HDLs with D-4F, an apoA-I mimetic peptide able to bind oxidized fatty acids, restores the antioxidant capacity of T2DM HDLs [81].

### 4.4. Nitric Oxide Production

The production of nitric oxide (NO) is important for normal endothelial function and protects against endothelial dysfunction, an early hallmark of atherosclerosis. HDLs improve the bioavailability of NO in vasculature mainly by inducing the activating-phosphorylation of eNOS at serine 1177, which then promotes NO synthesis. NO production contributes to a number HDL’s beneficial effects, such as vasorelaxation, inhibition of NF-κB pathway, and endothelium repair [162]. Several mechanisms are involved in eNOS activation mediated by HDLs. These include the binding of HDL-apoA-I to SR-B1 [163] and ABCG1 [164], the binding of HDL-S1P to S1PR1/3 receptors [165], the activation of mitogen-activated protein (MAP) kinases [166], the inhibition of protein kinase C (PKC) βII [148], cholesterol efflux facilitating the dissociation of eNOS from caveolin-1 [164], and, lastly, the suppression of ROS production which preserves from eNOS uncoupling.

An overview of NO-mediated HDL functions in T2DM and MetS is presented in Figure 4. HDLs from T2DM patients have been demonstrated to be less able to induce the activating-phosphorylation of eNOS at serine 1177 [54], NO production [80], and vessel relaxation [80]. Interestingly, eNOS phosphorylation and activity are even affected in obese patients in the absence of diabetes [61].

HDL-S1P is important for HDL-induced eNOS activation, and it has been reported that the S1P depletion observed in HDLs from non-diabetic obese patients plays a significant role in their loss of ability to activate eNOS, unlike TG enrichment [61]. The preponderant role of HDL-S1P reduction was also demonstrated with HDLs from patients with CAD [167]. However, S1P depletion in HDLs is not observed in T2DM patients, unlike non-diabetic MetS individuals, suggesting the implication of factors other than HDL-S1P in this defect in T2DM.

The oxidation of HDLs could also play a role, since in vitro exposure of HDLs with MPO-derived oxidants decreases their ability to activate eNOS [80]. Oxidized HDLs are able to bind to LOX-1 receptors, triggering the activation of PKCβII, which in turn impairs NO production [148]. The decrease in HDL-PON1 activity may contribute to this phenomenon [148].

In addition, it was recently shown that the enrichment of HDLs with LCAT or lysoPCs enhances NO production in endothelial cells [168]. However, because HDL-lysoPCs were most often increased in T2DM [19,40] and MetS [46], such a mechanism is unlikely. Data on LCAT activity in diabetes and MetS are scarce, making it difficult to understand the role of LCAT in the loss of ability of HDLs to induce NO production.

### 4.5. Antiapoptotic Properties and Endothelium Repair

The integrity of endothelial cells is crucial for vascular homeostasis, and endothelial cell death triggers vascular damage and promotes inflammation and endothelial dysfunction. HDL particles can inhibit apoptosis in endothelial cells, thus preserving endothelium integrity [52,169,170].

The antiapoptotic activity of small HDL particles is reduced in MetS individuals [171,172], and is closely associated with altered physicochemical properties, such as core CE depletion and TG enrichment in apoA-I-poor HDL3c [171].

The depletion of HDLs in plasmalogens in T2DM and MetS may also play a role, since enrichment of rHDLs with plasmalogens enhances their antiapoptotic activity [52]. Changes in the HDL-apoM/S1P axis are also likely to be involved considering its important role in HDL antiapoptotic activity [169,173].

Lastly, the in vivo capacity of early endothelial progenitor cells (EPCs) from T2DM patients to repair endothelial cells is severely reduced compared to healthy subjects [80], and the number of circulating EPCs is inversely associated with diabetes [174]. HDLs may promote EPC-mediated endothelium repair via induction of NO production, and HDLs from T2DM stimulate the production of NO in EPCs less [81]. The decrease in HDL-PON1 activity may also contribute to this defect, since its inhibition ex vivo reduces endothelium repair after carotid artery injury in mice [148].

## 5. Antidiabetic Properties

Emerging data suggest that HDL particles could actually contribute to the development of diabetes. Firstly, HDL-C levels are inversely associated with T2DM development in epidemiological studies [175,176,177], and this metric has been included in scores for T2DM risk [177,178]. In addition, Mendelian randomization studies showed that HDL-C elevation is associated with a lower risk of developing T2DM [179,180]. Moreover, both HDL particle size and the concentration of large HDL particles were inversely associated with incident T2DM in the general population [21]. Moreover, CETP inhibitors, which increased HDL-C concentration of 29 to 132% in large interventional studies, reduced the risk of new-onset diabetes by 16% on average [181].

Many of the antidiabetic mechanisms induced by HDLs or apoA-I have now been identified (recently reviewed in [182]). Infusions of rHDLs and apoA-I stimulate insulin secretion and reduce plasma glucose concentrations in obese mice and in T2DM patients [183,184]. From a mechanistic point of view, apoA-I enhances the expression of key enzymes involved in insulin maturation in β-cells [185]. In addition, HDLs protect β-cells from apoptosis induced by endoplasmic reticulum stressors [186].

Moreover, HDLs and apoA-I increase glucose uptake in skeletal muscle cells independently of the stimulation of insulin secretion [184]. First, they activate the insulin-independent AMP-activated protein kinase (AMPK) pathway [184], and apoA-I also stimulates the insulin receptor pathway [187]. Ultimately, this promotes the expression of glucose transporter type 4 (GLUT4) at the surface of the cell [187].

## 6. Conclusions

This review highlights the key findings regarding HDL alterations in T2DM and obesity, providing a better understanding of their potential role in CV risk. HDL metabolism is significantly changed in individuals with these conditions with an accelerated catabolism. HDLs are smaller and largely modified regarding their content in proteins and lipids, mainly marked by an enrichment in TG, a depletion in many proteins and glycoxidative modifications. Such alterations likely contribute to some functional defects of HDLs. Although the exact ability of HDLs to promote cholesterol efflux remains a matter of debate in insulin-resistant conditions, and several of the other atheroprotective functions of HDLs are impaired. Several studies showed that HDLs are less able to inhibit the NF-κB pro-inflammatory pathway and, subsequently, the adhesion of monocytes on endothelium and their recruitment into the subendothelial space. In addition, the antioxidative function of HDL particles is diminished, thus facilitating the deleterious effects of oxidized low-density lipoproteins on vasculature. Lastly, the HDL-induced activation of eNOS is less effective in T2DM and in MetS, which contributes to several HDL functional defects such as an impaired capacity to promote vasodilatation and endothelium repair, or to counteract ROS production and inflammation.

HDL-induced CEC and downregulation of VCAM-1 both predicted CV events in large prospective studies, independently of HDL-C [106,107,139]. In addition, HDL CEC is more strongly and inversely associated with incident CV events than circulating levels of HDL-C [106,107]. This highlights the potential interest of a biomarker based on HDL functionality, which could be used to better assess CV risk in clinical practice. However, the disadvantages of cell culture systems, such as handling time or lack of reproducibility, as well as isolation of HDL fractions, are undoubtedly major obstacles to the use of such procedures in clinical laboratories. A simple and easy to use test is required for wide use in the routine. The circulating number of HDL particles assessed by nuclear magnetic resonance spectroscopy appears as an interesting candidate, since it can predict CV events [188,189]. In addition, Murakami et al. have recently developed a promising cell-free assay, which evaluates cholesterol uptake capacity of whole serum using an automated immunoassay [190]. Further prospective studies are needed to evaluate the potential benefit of such surrogates (versus traditional CV risk factors) for predicting CV events, and whether they perform as well as cell-based functional tests. The perspective of validating a simple test that reflects HDL function will undoubtedly be an exciting area of study in the coming years.

Otherwise, the functional alterations of HDLs are interesting endpoints for interventional studies conducted in a context of T2DM or obesity. As far as lifestyle changes are concerned, physical activity significantly improves HDL functions in MetS patients [172]. Similarly, a reduction in calorie intake associated with physical activity improves antioxidative HDL3c function in MetS patients [191]. In addition, smoking cessation ameliorates HDL-mediated CEC and the antioxidative properties of HDL [192,193].

Concerning glucose-lowering agents, little is currently known about the effects of metformin, the first-line treatment of T2DM, on HDL function. In vitro pre-incubation of glycated HDLs with metformin decreases their content in AGEs and restores their ability to promote cholesterol efflux [126,194]. In an indirect way, metformin also improves HDL-mediated CEC by upregulating the expression of ABCA1 and ABCG1 transporters in macrophages [195,196]. Regarding thiazolidinediones, our group has observed no improvement in the vasorelaxant effect of HDLs in T2DM individuals treated with rosiglitazone or pioglitazone, despite an increase in HDL-C [197]. Data on the impact of glucagon-like peptide-1 (GLP-1) receptor agonists or dipeptidyl peptidase-4 inhibitors on HDL functions are also very scarce. The GLP-1 receptor agonist liraglutide restores endothelium-protective properties of HDL in diet-induced obese rats [198]. Lastly, regarding sodium-glucose cotransporter type 2 inhibitors, dapaglifozin has no impact on CEC after adjustment for age and BMI [199].

Regarding the effects of lipid-lowering agents on HDL kinetics and functions, rosuvastatin corrects HDL-apoA-I kinetics abnormalities, with an effect on both hypercatabolism and increased production rate in T2DM patients [39] and in non-diabetic patients with MetS [200]. However, atorvastatin does not modify HDL-apoA-I kinetics in obese individuals [201]. For fibrates, it has been shown that bezafibrate has no effect on anti-inflammatory and antioxidant HDL functions [202], and conflicting results were reported for CEC in T2DM patients [203]. Proprotein convertase subtilisin/kexin type 9 inhibitors improve HDL CEC mediated by ABCG1 and aqueous diffusion in patients with familial hypercholesterolemia [204], but no data exist specifically in T2DM or obesity to our knowledge. Anacetrapib, the only CETP inhibitor that reduces CHD in large trials, ameliorates CEC including after adjustment for HDL-C level changes [205]. However, the CETP inhibitor evacetrapib also increases CEC [206], although it failed to demonstrate beneficial CV effects [207].

Finally, bariatric surgery appears to be beneficial for HDL functions. It improves HDL-mediated CEC in obese [198,208,209] and T2DM patients [210], increases antioxidative HDL function in obese individuals [211], stimulates HDL-induced NO production by endothelial cells [198], and restores HDL-mediated inhibition of endothelial cell apoptosis and inflammation [198].

## Figures and Tables

**Figure 1 metabolites-13-00253-f001:**
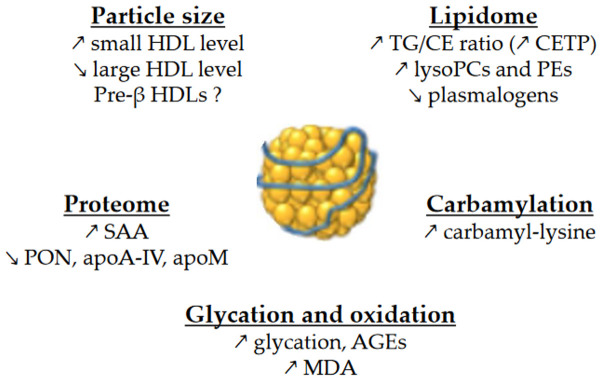
Main changes in the size and composition of HDLs in T2DM and MetS. ↗ and ↘ mean increase and decrease, respectively. AGEs, advanced glycation end-products; CE, cholesteryl esters; CETP, cholesteryl ester transfer protein; MDA, malondialdehyde; PCs, phosphatidylcholines; PEs, phosphatidylethanolamines; PON, paraoxonase; SAA, serum amyloid A; TG, triglycerides.

**Figure 2 metabolites-13-00253-f002:**
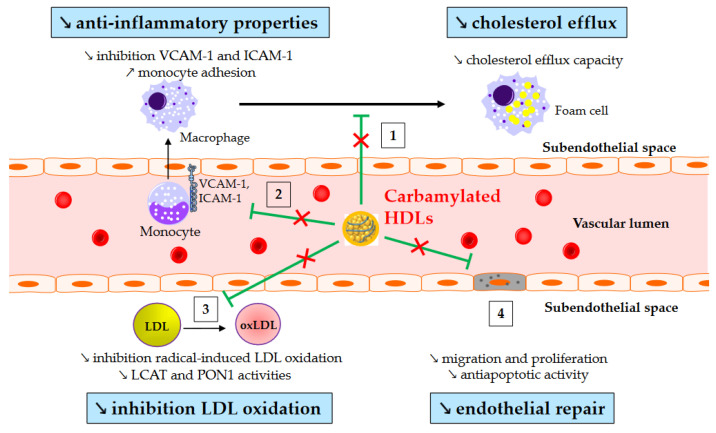
Impaired atheroprotective functions of carbamylated HDLs. ↗ and ↘ mean increase and decrease, respectively. Green arrows represent the functions of healthy HDLs. (1) Carbamylated HDLs partially lose their ability to remove cholesterol from macrophages, and (2) to inhibit monocyte adhesion and recruitment into the subendothelial space. (3) Carbamylated HDLs are less able to protect LDLs from oxidation, likely due to reduced lecithin-cholesterol acyltransferase (LCAT) and paraoxonase-1 (PON1) activity. (4) Lastly, carbamylated HDLs have an impaired capacity to facilitate endothelial repair.

**Figure 3 metabolites-13-00253-f003:**
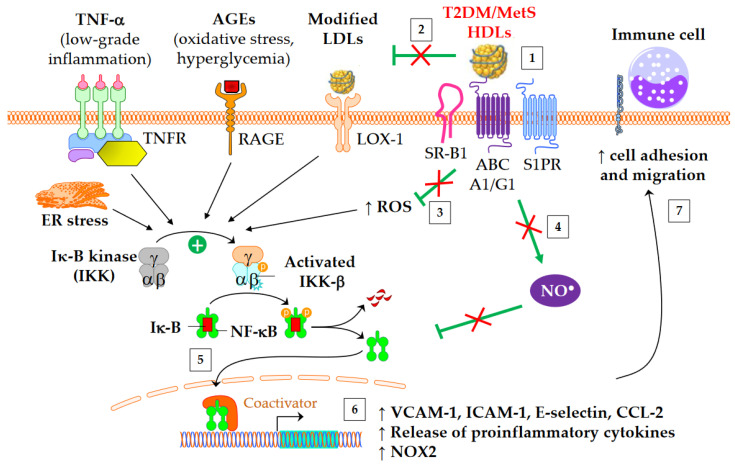
Anti-inflammatory functions of HDLs in T2DM and MetS. ↑ means increase. The inflammatory NF-κB pathway is triggered by several mediators, including TNF-α, advanced glycation end-products (AGEs), modified LDLs, reactive oxygen species (ROS), and endoplasmic reticulum (ER) stress. This leads to an increased gene expression of adhesion molecules, proinflammatory cytokines, and NADPH oxidase (NOX2). Green arrows represent the functions of healthy HDLs. (1) The binding of HDLs to receptors is modified by conformational changes in HDL particles in insulin resistant conditions. The depletion in S1P of MetS HDLs likely decreases the binding to S1P receptors (S1PR). (2) HDL-mediated protection of LDLs against oxidation is affected, (3) in particular due to the loss of capacity of HDLs to dampen ROS production. This promotes the NF-κB activation triggered by the recognition of oxidized LDLs by scavenger receptors in vasculature, particularly by LOX-1 (i.e., SR-E1) and SR-A1. (4) The activation of endothelial NO synthase by HDLs is reduced (see 4.3) and subsequently inhibits nitrosylation of NF-κB. (5) Ultimately, HDLs are less able to inhibit the translocation of NF-κB into the nucleus, and, (6) afterwards, the gene expression of adhesion molecules and proinflammatory cytokines. (7) This facilitates the recruitment of immune cells into the subendothelial space.

**Figure 4 metabolites-13-00253-f004:**
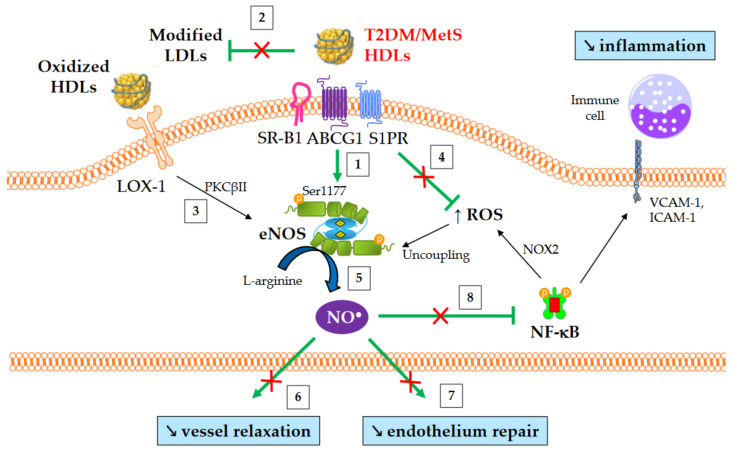
NO-mediated HDL functions in T2DM and MetS. ↑ and ↘ mean increase and decrease, respectively. Green arrows represent the functions of healthy HDLs. (1) HDLs from T2DM and non-diabetic MetS individuals are less able to induce Akt-dependent eNOS phosphorylation at Ser1177. (2) HDL-mediated protection of LDLs against oxidation is affected, thus facilitating eNOS inhibition after the binding of modified LDLs to LOX-1. (3) Oxidized HDLs are also able to bind to LOX-1 receptor, leading to PKCβII activation and subsequently to eNOS inhibition. (4) The loss of capacity of HDLs to dampen ROS production promotes eNOS uncoupling. (5) Ultimately, HDL-mediated NO production is reduced, (6) affecting the relaxation of vascular smooth muscle cells and (7) endothelium repair. (8) Reduced NO synthesis diminishes the inhibition of nitrosylation of NF-κB, as well as NOX2-mediated ROS production and the recruitment of immune cells into the subendothelial space.

**Table 1 metabolites-13-00253-t001:** Overview of CEC studies in T2DM and/or MetS.

Study	Subjects ^1^	Cell Model	Acceptor	CEC in Patients
Alenezi [120]	22 MetS/9 controls Glucose: 5.7 ± 1.4 mmol/L TG: 3.84 ± 1.77 mmol/L	Human fibroblasts	Purified apoA-I	No difference in cholesterol and phospholipids efflux
Annema [121]	297 MetS/255 non-MetS HbA_1c_: 6.2 ± 0.9% TG: 1.90 (1.40–2.20) mmol/L	Human THP-1 macrophages	ApoB-depleted serum	No difference for T2DM 7% decrease in MetS patients
Apro [18]	35 T2DM/35 controls	Human THP-1 macrophages	HDLs from plasma and interstitial fluid	10% decrease (plasma) 28% decrease (interstitial fluid)
Cavallero [112]	14 T2DM/12 controls HbA_1c_: 5.1 ± 0.9% TG: 2.05 ± 0.73 mmol/L	Ob 1771 preadipocyte	LpA-I ^2^	50% decrease
Denimal [41]	20 T2DM/25 controls HbA_1c_: 10.0 ± 2.3% TG: 2.38 ± 1.01 mmol/L	Human THP-1 macrophages	ApoB-depleted serum	No difference
Dullaart [28]	76 MetS/94 controls Glucose: 8.4 ± 2.6 mmol/L TG: 1.35 mmol/L	Human fibroblasts	Whole plasma	3.5% increase
Dullaart [30]	75 T2DM/75 controls HbA_1c_: 6.7 ± 1.0% TG: 1.73 (1.17–2.17) mmol/L	Human fibroblasts	Whole plasma	No difference
Feng [81]	6 T2DM/6 controls HbA_1c_: 10.9 ± 1.3% TG: 2.2 ± 0.8 mmol/L	Human THP-1 macrophages	HDLs	No difference
He [117]	19 T2DM/20 controls HbA_1c_: 7.2 ± 1.2% TG: 1.47 ± 0.44 mmol/L	Baby hamster kidney (BHK) cells	Fractioned HDLs	23% decrease in ABCA1-mediated CEC of small HDLs
Kashyap [97]	9 T2DM/8 controls HbA_1c_: 6.3 ± 0.3% TG: 1.13 ± 0.42 mmol/L	Murine RAW264.7 macrophages	ApoB-depleted serum	Decrease in ABCA1-mediated and total CEC
Low [27]	26 T2DM/26 controls HbA_1c_: 7.7 ± 1.2% TG: 1.7 ± 0.8 mmol/L	Human THP-1 macrophages	Whole plasma, apoB-depleted serum, HDLs	30% increase (whole plasma), 34% increase (apoB-depleted serum), 50% increase (HDLs)
Lucero [122]	35 MetS/15 controls Glucose: 7.4 ± 3.3 mmol/L TG: 1.73 ± 0.62 mmol/L	ABCA1 or ABCG1- transfected BHK cells	ApoB-depleted serum	19% increase in ABCA1-mediated CEC
Murakami [116]	36 T2DM/9 controls HbA_1c_: 9.5 ± 1.7% TG: 1.49 ± 0.77 mmol/L	Human THP-1 macrophages	HDLs	7.3% decrease in CEC
Passarelli [115]	18 T2DM/26 controls HbA_1c_: 12 ± 2% TG: 2.64 ± 1.51 mmol/L	Murine peritoneal macrophages	HDL subfractions	>50% decrease in HDL3-mediated CEC
Shiu [34]	172 T2DM/175 controls HbA_1c_: 8.2 ± 1.4% TG: 1.40 (0.90–2.10) mmol/L	Fu5AH hepatoma cells	Whole serum	9% decrease in SR-B1-mediated CEC. No difference in ABCA1-mediated CEC.
Shiu [36]	180 T2DM/120 controls	Fu5AH hepatoma cells	Whole serum	17% decrease in ABCA1-mediated CEC
Syvänne [113]	100 T2DM/81 controls TG: 2.15 ± 1.20 mmol/L	Fu5AH hepatoma cells	Whole plasma	Reduction in CEC
Tsun [68]	264 T2DM/275 controls HbA_1c_: 8.5 ± 1.7% TG: 1.60 (1.2–2.2) mmol/L	Fu5AH ABCG1-transfected CHO cells	Whole serum	Reduction in ABCG1- and SR-B1- mediated CEC
Yassine [111]	45 T2DM with hypertriglyceridemia/ 26 T2DM without hypertriglyceridemia HbA_1c_: 8 ± 2%/8 ± 2% TG: 2.55 ± 0.99/1.03 ± 0.25 mmol/L	BHK	ApoB-depleted serum	14% increase in ABCA1-dependent CEC in T2DM patients with hypertriglyceridemia
Zhou [37]	60 T2DM/20 controls HbA_1c_: 8.3 ± 0.3% TG: 1.4 (1.0–2.6) mmol/L	Fu5AH hepatoma cells	Whole serum	20 and 14% reduction in ABCA1-mediated and SR-B1-mediated CEC, respectively
Zhou [114]	137 T2DM/75 controls HbA_1c_: 8.0 ± 1.3% TG: 1.50 (1.00–1.90) mmol/L	Fu5AH hepatoma cells	Whole serum	7% reduction

^1^ Data on glycemic control (fasting glycemia or HbA_1c_) and triglyceridemia (TG) in patients with type 2 diabetes (T2DM) or metabolic syndrome (MetS) are reported, if available. Values are reported as means ± standard deviations or means (1st–3rd quartiles).^2^ LpA-I: HDL particles containing apoA-I but not apoA-II.

**Table 2 metabolites-13-00253-t002:** Overview of studies on anti-inflammatory properties of HDLs in patients with T2DM.

Study	Subjects ^1^	Cell Model	Results in Patients
Chen [87]	102 T2DM with CAD/46 T2DM without CAD/40 controls HbA_1c_: 7.7 ± 1.3% TG: 1.77 ± 1.15 mmol/L	HUVECs and monocytes	Increased adhesion of monocytes to HUVECs in T2DM patients with CAD compared to controls.
Denimal [41]	20 T2DM/25 controls HbA_1c_: 10.0 ± 2.3% TG: 2.38 ± 1.01 mmol/L	HUVECs	No difference in the ability of T2DM HDLs to inhibit VCAM-1 gene expression compared to control HDLs
Ebtehaj [142]	40 T2DM/36 controls HbA_1c_: 6.7 ± 2.9% TG: 1.67 (1.22–2.16) mmol/L	HUVECs	Decreased ability of T2DM HDLs to inhibit VCAM-1 gene expression
Liu [146]	6 T2DM/6 controls HbA_1c_: 10.8 ± 1.12% TG: 2.05 ± 0.21 mmol/L	Human THP-1 macrophages	Decreased ability of T2DM apoA-I to inhibit the LPS-induced release of TNF-α and IL-1β
Liu [73]	6 T2DM/6 controls HbA_1c_: 10.8 ± 1.12% TG: 2.05 ± 0.21 mmol/L	Human THP-1 macrophages	Decreased ability of T2DM apoA-I to inhibit the release of TNF-α and IL-1β
Morgantini [140]	93 T2DM/31 controls HbA1c: 8 ± 2% TG: 1.93 ± 1.54 mmol/L	HAECs and monocytes	Reduction in ability of T2DM HDLs to inhibit LDL-induced migration of monocytes

CAD, coronary artery disease; HAECs, human aortic endothelial cells; HUVECs, human umbilical vein endothelial cells; LPS, lipopolysaccharide; VCAM-1, vascular adhesion molecule-1; TG, triglyceridemia. ^1^ Data on glycemic control (fasting glycemia or HbA_1c_) and triglyceridemia (TG) in patients with T2DM or MetS are reported, if available. Values are reported as means ± standard deviations or means (1st–3rd quartiles).

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
