# Peer review of "High-Density Lipoprotein Alterations in Type 2 Diabetes and Obesity"

_metabolites, 2023, doi:10.3390/metabo13020253_

Round 1

Reviewer 1 Report

This is a very thoroughly researched, well written, and timely article on HDL and its alterations in Diabetes and Obesity.

Author Response

“This is a very thoroughly researched, well written, and timely article on HDL and its alterations in Diabetes and Obesity.”

We would like to thank the Reviewer for his positive feedback on our manuscript. We submitted our revised manuscript to an English-speaking colleague for proofreading (added in the Acknowledgments section). As a result, several changes have been made throughout the manuscript, which are marked up using the “Track Changes” function, as required by the Editor.

Reviewer 2 Report

The author presented a research work in this article in which they found that alterations in HDL proteins are responsible for the pathogenesis and development of DM and its associated disorder i.e., obesity. The manuscript needs to be improved regarding the following aspects.

      The text is quite confusing. The introduction could be remodulated in a more flowing way and without skipping a topic and then returning to it.

      Backgrounds of Diabetes mellitus in the introduction section have not discussed as it is very important that there are several factors that are responsible for the development and pathogenesis of DM and its associated disorders. It is important to highlight the pathogenesis and risk factors associated with diabetes. I also recommend that authors should also discuss some biomarkers that are helpful to predict the development of DM. Notably genetics, oxidative stress, retinopathy, neuropathy, etc. are some of the important factors.

·       In Figure 1,2 , 3 and 4 : Explain what is meant by these arrow "        " and

" ". Or the directions these arrows should be explained in the figure legends part.

·       References used should be updated, where it is noted that there were references from 1996 -1999 update these references and all other references from 2017 onwards.

      Future prospective of study is not mentioned.

      At least one illustrative figure may be provided to highlight the summary of this study.

      The key findings described in the results and discussion sections are quite interesting, but they do not corroborate with the conclusion of this study. I suggest the authors to revise the conclusion section.

      Abbreviations are missing.

      There are many formatting mistakes.

      There are some grammatical errors in this manuscript, such as verbs and prepositions. The manuscript needs extensive review by an author.

      Insert the correct format style for journals in the references in the text and references list.

      Resolution and colors of all figures should be improved.

      Update table 1 by adding more information.

      In terms of the presentation, I suggest that the previous studies in the literature should be expanded with a table to make it easier to compare the conclusions from such work.

Author Response

“The author presented a research work in this article in which they found that alterations in HDL proteins are responsible for the pathogenesis and development of DM and its associated disorder i.e., obesity. The manuscript needs to be improved regarding the following aspects.”

We would like to thank the Reviewer for these valuable comments, which have helped us to improve the manuscript. All revisions made to the manuscript are marked up using the “Track Changes” function, as required by the Editor.

  1. “The text is quite confusing. The introduction could be remodulated in a more flowing way and without skipping a topic and then returning to it.”

We therefore moved the last paragraph of the introduction (in the initial manuscript) to avoid the back and forth between themes.

  1. “Backgrounds of Diabetes mellitus in the introduction section have not discussed as it is very important that there are several factors that are responsible for the development and pathogenesis of DM and its associated disorders. It is important to highlight the pathogenesis and risk factors associated with diabetes. I also recommend that authors should also discuss some biomarkers that are helpful to predict the development of DM. Notably genetics, oxidative stress, retinopathy, neuropathy, etc. are some of the important factors.”

We therefore inserted a paragraph at the beginning of the manuscript to summarize the risk factors for developing type 2 diabetes. We also introduced the notion that certain biomarkers can predict the development of type 2 diabetes, and mentioned a review on this topic. We also inserted a sentence in the second paragraph of the revised manuscript to discuss the complex pathogenesis of diabetes and its complications, and thus better introduce diabetic dyslipidemia.

  1. “In Figure 1, 2, 3 and 4: Explain what is meant by these arrow " " and " ". Or the directions these arrows should be explained in the figure legends part.”

We have clarified the meaning of the arrows in the figure legends.

  1. “References used should be updated, where it is noted that there were references from 1996 -1999 update these references and all other references from 2017 onwards.”

We therefore substituted the references (in the initial manuscript) #7 (Vergès et al. Diabetologia 2015), #50 (Kiziltunç et al. Clin. Biochem. 1997), #97 (Golay et al. J. Clin. Endocrinol. Metab. 1987), #100 (Pietzsch et al. Diabetes 1998), #141 (Navab et al. J. Clin. Invest. 1991), #142 (Cockerill et al. Arterioscler. Thromb. Vasc. Biol. 1995), #147 (Ashby et al. Arterioscler. Thromb. Vasc. Biol. 1998) and #166 (Abbott et al. Arterioscler. Thromb. Vasc. Biol. 1995). We maintained the references #112 (1995) and 113 (1996) (in the revised manuscript) because we wanted to be as exhaustive as possible for reporting studies on cholesterol efflux capacity of HDLs (Table 1).

  1. “Future prospective of study is not mentioned.”

As a future prospective, we highlighted in the discussion of the revised manuscript the potential interest of a biomarker based on HDL functionality, which could be used to better assess cardiovascular risk in clinical practice. We also discussed the difficulties of such a test in clinical laboratories, and we then introduced the potential interest of surrogates such as the number of HDL particles.

  1. “At least one illustrative figure may be provided to highlight the summary of this study.”

As proposed in the instructions for authors, we added a graphical abstract in the revised manuscript in order to highlight the summary of this study. We followed the guidelines to edit this graphical abstract. We inserted the graphical abstract just after the abstract in the revised manuscript (as it appears online), and it is also transmitted to the editorial office in a separate file.

  1. “The key findings described in the results and discussion sections are quite interesting, but they do not corroborate with the conclusion of this study. I suggest the authors to revise the conclusion section.”

As suggested by the reviewer, we reinforced the first paragraph of the conclusion in order to better summarize the main findings.

  1. “Abbreviations are missing.”

We carefully checked if all abbreviations are well defined in parentheses the first time they appear in the abstract, main text, and in figure or table captions, as required in the instructions for authors. We therefore defined the following abbreviations in the revised manuscript: NF-kB (in the abstract), CV (in the introduction), CAD (page 3), FoxO (page 4), TNF (page 5), ABC (page 5), and MAP (page 13).

  1. “There are many formatting mistakes. There are some grammatical errors in this manuscript, such as verbs and prepositions. The manuscript needs extensive review by an author.”

We submitted our revised manuscript to a native English-speaking colleague for proofreading (added in the Acknowledgments section). As a result, several changes have been made throughout the manuscript, which are marked up using the “Track Changes” function.

  1. “Insert the correct format style for journals in the references in the text and references list.”

We therefore checked that the reference numbers are well placed in square brackets [ ] before the punctuation in the body text, as required in the instructions for authors. We also checked that we had used the recommended style for journal articles (available on https://www.mdpi.com/authors/references), i.e. “8. Bowman, C.M.; Landee, F.A.; Reslock, M.A. Chemically Oriented Storage and Retrieval System. 1. Storage and Verification of Structural Information. J. Chem. Doc. 1967, 7, 43-47; DOI:10.1021/c160024a013.”. In the revised manuscript, we therefore added “.” in journal abbreviations. We used the reference generator available on the website during the submission process.

  1. “Resolution and colors of all figures should be improved.”

We improved the resolution and colors of all figures included in the main document. Following the instructions for authors, all original figures were also transmitted to the Editorial office during the submission in addition to the main document.

  1. “Update table 1 by adding more information.”

We therefore added in the revised Table 1 a column entitled “subjects”, which includes the main data on metabolic control in T2DM/MetS patients, such as HbA1c and fasting glycemia and triglyceridemia (when available).

  1. “In terms of the presentation, I suggest that the previous studies in the literature should be expanded with a table to make it easier to compare the conclusions from such work.”

We therefore added the Table 2 in the revised manuscript, which summarizes studies on anti-inflammatory properties of HDLs in patients with T2DM. Consistent with our answer to the previous comment, we also reported data on metabolic control, i.e. HbA1c and fasting triglyceridemia in T2DM patients.

Reviewer 3 Report

The authors aimed to discuss the main findings regarding qualitative and quantitative alterations of HDL in T2DM and obesity in the context of cardiovascular diseases.

This is a well organized and presented, nicely illustrated review that summarize all the sufficient data on qualitative and quantitative alterations of HDL in T2DM and obesity.

Comments:

1.     Difficulties of HDL function measurements in the everyday clinical practice could be discussed.  

2.     English need some editing.

Author Response

“The authors aimed to discuss the main findings regarding qualitative and quantitative alterations of HDL in T2DM and obesity in the context of cardiovascular diseases.

This is a well-organized and presented, nicely illustrated review that summarize all the sufficient data on qualitative and quantitative alterations of HDL in T2DM and obesity.”

We would like to thank the reviewer for these positive comments on our manuscript. All revisions made to the manuscript are marked up using the “Track Changes” function, as required by the Editor.

  1. “Difficulties of HDL function measurements in the everyday clinical practice could be discussed.”

We fully agree and have therefore inserted a paragraph about this concern in the conclusion section.   

  1. “English need some editing.”

We submitted our revised manuscript to a native English-speaking colleague for proofreading (added in the Acknowledgments section). As a result, several changes have been made throughout the manuscript, which are marked up using the “Track Changes” function.